# Design and Synthesis of New Hydantoin Acetanilide Derivatives as Anti-NSCLC Targeting EGFR^L858R/T790M^ Mutations

**DOI:** 10.3390/ph15070857

**Published:** 2022-07-12

**Authors:** Moamen A. Hassanin, Muhamad Mustafa, Mohammed A. S. Abourehab, Heba A. Hassan, Omar M. Aly, Eman A. M. Beshr

**Affiliations:** 1Department of Medicinal Chemistry, Faculty of Pharmacy, Minia University, Minia 61519, Egypt; moamenawad15@gmail.com (M.A.H.); hebahassan2009@live.com (H.A.H.); emanbeshr@yahoo.com (E.A.M.B.); 2Department of Medicinal Chemistry, Faculty of Pharmacy, Deraya University, Minia 61111, Egypt; 3Department of Pharmaceutics, Faculty of Pharmacy, Umm Al-Qura University, Makkah 21955, Saudi Arabia; maabourehab@uqu.edu.sa; 4Department of Medicinal Chemistry, Faculty of Pharmacy, Port Said University, Port Said 42511, Egypt

**Keywords:** EGFR, mutation, hydantoin, anticancer, molecular docking

## Abstract

Epidermal Growth Factor Receptor (EGFR), its wild type and mutations L858R/T790M, is overexpressed in non-small cell lung cancer (NSCLC) patients and is considered an inevitable oncology target. However, while the potential EGFR inhibitors have been represented in the literature, their cellular activity failed to establish broad potency against EGFR and its mutations. This study identifies a new series of EGFR^L858R/T790M^ inhibitors bearing hydantoin acetanilides. Most compounds revealed strong antiproliferative activity in a range of NSCL cancer models (A549, H1975, and PC9), in which **5a** and **5f** were the most potent. Compounds **5a** and **5f** possessed potent anticancer activity on H1975 cells with IC_50_ values of 1.94 and 1.38 µM, respectively, compared to 9.70 µM for erlotinib. Favorably, **5a** and **5f** showed low activity on WI-38 normal cells. Western blotting and an EGFR kinase assay test proved the significant EGFR inhibitory activity of **5a**. Besides, active hydantoin derivative **5a** strongly arrested the cell cycle at the sub G1 and S phases and triggered apoptosis in A549 cells. These results imply that **5a** could be considered a promising lead compound for additional development as a potential active agent for anticancer therapy.

## 1. Introduction

The Epidermal Growth Factor family (EGFR) is a transmembrane protein that plays a critical role in cell proliferation, survival, migration, adhesion, and differentiation [1]. Unfortunately, overexpression of the EGFR gene initiates diverse downstream signaling pathways, resulting in cancer aggressiveness and invasiveness [2]. Non-small cell lung cancer (NSCLC) is one of the major causes of cancer-related mortality and morbidity globally, accounting for about 80–90% of all lung malignancies [2,3,4,5]. EGFR overexpression and mutation (L858R and T790M) is highly prevalent in NSCLC [6,7]. Approximately 10–50% of NSCLC patients harbor EGFR containing the activating mutations [8]. Therefore, EGFR is a valuable clinically proven therapeutic target for anticancer therapy, particularly in the treatment of NSCLC.

The approved first-generation EGFR tyrosine kinase inhibitors, such as gefitinib and erlotinib (Figure 1), provided significant clinical benefits in patients with activating mutations, resulting in marked tumor shrinkage [9,10,11]. However, acquired drug resistance usually develops after approximately a year of gefitinib or erlotinib treatment [4]. According to reports, around 60% of EGFR-mutant NSCLC patients have the EGFR^T790M^ mutation, which renders first-generation EGFR inhibitors less effective in these patients [1]. Drastically, the T790M mutation boosts the affinity of the EGFR receptor for ATP, leading to a lowering of the ability of EGFR inhibitors to compete with ATP effectively [12]. In the T790M mutation, a secondary Thr790 to Met790 mutation occurs in the gatekeeper site of the EGFR catalytic domain, whereas the L858R mutation is a single-nucleotide substitution that replaces leucine with arginine at position 858 [3,4]. In the case of the latter mutation, the EGFR receptor shows a 50-fold of raised catalytic activity, symbolizing a vital hub for cancer cells in rising cell growth, downstream signaling, and metastasis [13]. As a result, the binding of these inhibitors is sterically blocked by the bulky methionine moiety, which disrupts water-mediated hydrogen bond formation between the inhibitors and the T790M of EGFR [4,12]. In this respect, second-generation inhibitors (afatinib and dacomitinib) have been developed to overcome the resistance associated with the T790M mutation (Figure 1) [14,15]. However, they have poor kinase selectivity between EGFR^T790M^ mutants and the wild type (WT) EGFR. In addition, these inhibitors are associated with dose-limiting toxicities [2,16]. Seminal work was conducted to discover the third-generation EGFR inhibitors, rociletinib and Osimertinib, that block both activating mutations and the T790M drug resistance mutation of EGFR (Figure 1) [17,18,19,20]. As a result, they have the benefits of a better selectivity profile, an increased time to resistance, and better toxicity profiles. Among these third-generation inhibitors, osimertinib was identified as a breakthrough drug for treating mutant NSCLC and was authorized by the FDA in 2015 to treat NSCLC patients with the T790M EGFR mutation. Subsequently, it is still important to discover more compounds that could target EGFR and its mutations. Moreover, hydantoin moiety is emerging as a cornerstone scaffold in potential anticancer agents, particularly EGFR inhibitors (Figure 2) [21,22,23]. 

As part of our ongoing efforts to discover new probes in our armamentarium for treating cancer, a new series of hydantoin acetanilide derivatives were designed and synthesized to conquer NSCLC through inhibiting EGFR^WT^ and its mutations EGFR^L858R/T790M^. The prepared derivatives **5a**–**5n** were evaluated for their anti-NSCLC activity in A549, H1975, and PC9 cell lines. The most potent members **5a** and **5f** were tested for their safety on the normal fibroblast WI-38 cells, and their EGFR^WT/L858R/T790M^ inhibitory activity. The effect of **5a** on the EGFR downstream signaling transduction pathways Akt and ERK was investigated. Molecular docking was performed within the EGFR^T790M^ binding site to gain further insights into possible modes of action.

### Rational Design of EGFR Inhibitors

Designing an inhibitor with a higher affinity to Met790 in NSCLC is our aim to overcome the resistance associated with the T790M mutation in which Met790 replaces Thr790 (Figure 3). The hydantoin group was designed to interact with Met790 in a manner similar to ribose of the agonist ATP, whereas the phenyl ring needs to interact with Leu777 and Asp855 similar to the adenine group of the ATP. The carbonyl group of the hydantoin group interacts with Thr854 similar to that of gefitinib. The carbonyl of the acetamide group binds with Lys745 by hydrogen bonding similar to ATP but ATP interacts with Lys745 by an ionic bond. The acetamide N-substituted phenyl group showed hydrophobic bonding with Cys797 of the kinase hinge area. The design of the target compounds shows the advantage of combining the binding properties of both ATP and gefitinib.

## 2. Results and Discussion

### 2.1. Chemistry

The synthesis of the alkylated hydantoin derivatives **5a**–**5n** (Table 1) is outlined in Figure 1. The anilines **1a**–**1n** were treated with chloroacetyl chloride in the presence of triethylamine (TEA) to afford the corresponding acetylated anilines **2a**–**2n**. The parent hydantoin (5-methyl-5-phenylimidazolidine-2,4-dione) **4** was prepared in a good yield, via the non-stereospecific Bucherer–Bergs reaction, by refluxing acetophenone **3** with potassium cyanide and anhydrous ammonium carbonate [24]. In the presence of potassium carbonate and potassium iodide, hydantoin **4** was alkylated with the appropriate acylated aniline **2a**–**2n** to afford the targeted alkylated hydantoin derivatives **5a**–**5n**. The synthesized compounds were identified by ^1^HNMR, ^13^CNMR, and mass spectroscopy, and further confirmed by elemental analyses. The purity of compound **5a** was checked by HPLC analysis using a gradient elution of methanol: water mixture at a rate of 1.0 mL/min leading to elution of **5a** as a single peak after 19.94 min at λmax of 254 nm (supp info). ^1^H NMR spectra of compounds **5a**–**5n** showed the hydantoin methyl protons as a singlet integrated for three protons at approximately δ 1.69–1.74 ppm. The hydantoin amidic NH proton appeared as a singlet at δ 9.03–8.92 ppm, whereas the other amide proton appeared more downfield shifted as a singlet at 10.71–9.48 ppm due to the anisotropic effect of the benzene ring.

### 2.2. In Vitro Antiproliferative Activity 

The synthesized hydantoin acetanilide derivatives **5a**–**5n** were evaluated for their anticancer activity in three various cells derived from NSCLC tumor subpanels, including A549 (harbor EGFR^WT^), H1975 (harbor EGFR^L858R/T790M^), and PC9 (harbor EGFR^L858R^) cells, using the MTT assay (Table 2). Erlotinib was used as a positive control. Interestingly, six hydantoin derivatives exhibited promising activity. Among them, **5a**, **5b**, **5e**, **5f**, **5i**, and **5n** were the most potent against the three evaluated cell lines (average IC_50_ = 2.57–9.29 µM). The synthesized compounds’ half-maximal inhibitory concentration (IC_50_) revealed a substantial ability of **5a**, **5e**, **5f**, **5i**, **5j**, **5l**, and **5n** to inhibit A549 lung cancer growth; their IC_50_ values were superior to erlotinib. On the other hand, compounds **5b**, **5h**, and **5m** possessed moderate potency (IC_50_ = 7.19–14.54 µM), whereas compounds **5c**, **5d**, **5g**, and **5k** showed a low potency (IC_50_ = 35.82–91.39 µM). Regarding H1975 cells, compounds **5a**, **5b**, **5e**, **5f**, **5g**, **5k**, and **5m** showed better activity than the reference erlotinib, in which **5a** and **5f** showed their activity in low micromolar concentrations. Interestingly, all the prepared compounds showed better anticancer activity in the PC9 cell line than the reference erlotinib, except **5c** and **5m**. These results revealed that the synthesized hydantoin acetanilide derivatives possessed promising anticancer potency, and among them **5a** and **5f** were the most potent. 

### 2.3. In Vitro EGFR Inhibition Assay

As our rationale was to target EGFR and its mutations, activities of the most potent derivatives **5a** and **5f** were evaluated against EGFR^WT^, EGFR^L858R^, and EGFR^T790M^ using EGFR kinase assay tests. Erlotinib was used as a positive control. The results in Table 3 showed that **5a** and **5f** have potent EGFR^WT^ inhibitory activity, however, they were approximately three and seven-folds lower than the reference erlotinib, respectively. On the other hand, **5a** showed approximately the same EGFR^L858R^ and EGFR^T790M^ inhibitory activities (IC_50_ = 0.05 and 0.09 µM, respectively) compared to erlotinib (IC_50_ = 0.03 and 0.07 µM, respectively), whereas **5f** showed a five and four-fold decrease in EGFR^L858R^ and EGFR^T790M^ inhibitory activities compared to erlotinib, respectively. These results revealed that **5a** was potent against the mutations of EGFR.

### 2.4. In Vitro Inhibition of the EGFR Downstream Pathway

The ability of **5a** to inhibit Akt and ERK, the downstream signaling transduction of EGFR, was investigated using Akt and ERK enzyme assays. Treating A549 cells with the IC_50_ value of **5a** resulted in a substantial inhibition of Akt and ERK (IC_50_ = 33.75 and 25.76 nM, respectively) (Table 4). Besides, **5a** revealed a comparable Akt and ERK inhibition compared to the reference erlotinib (IC_50_ = 25.41 and 17.46 nM, respectively). Interestingly, these findings indicated that the hydantoin derivative 5a targeted EGFR and inhibited its related downstream proteins ERK and Akt.

### 2.5. Inhibition of EGFR Phosphorylation Expression Using Western Blot Analysis

To further investigate the biochemical effects of the most active compound **5a** on EGFR, Western blot analysis was conducted on A549 cell lines. In addition, the obtained data were attuned to β-actin expression to exclude the deviations further. The hydantoin derivative **5a** maintained a strong EGFR inhibitory activity; treating A549 cells with 0.03 µM of **5a** significantly lessened the expression level of EGFR phosphorylation and showed a 2.5-fold decrease in the expression compared to the control A549 cells (Figure 4).

### 2.6. In Vitro Cytotoxic Activity towards Normal Fibroblast (WI-38) Cells

An MTT in vitro cytotoxic assay was conducted on the normal fibroblast WI-38 cells to examine the safety of **5a** and **5f**. Erlotinib was used as a positive control (Table 5). The obtained IC_50_ values revealed that both hydantoin derivatives possessed low cytotoxicity on the normal fetal lung WI-38 cells, in which **5a** and **5f** were 2.6 and 2.3-folds superior to the reference erlotinib, respectively. These results implied that **5a** and **5f** were barely cytotoxic on the normal WI-38 cell line.

### 2.7. Cell Cycle Analysis and Detection of Apoptosis

The active hydantoin derivative **5a** was further investigated for its effect on cell cycle progression and the induction of apoptosis in A549 cells. Exposure of A549 cells to **5a** at its IC_50_ value led to an interference with the normal cell cycle distribution of the cell (Figure 5). We observed that compound **5a** arrested the cell cycle at sub G1 and S phases (35.25% and 40.40%, respectively). Interestingly, **5a** caused a substantial boost of cells at pre-G1 by 22-folds compared to the control. Meanwhile, **5a** was evaluated for its ability to induce apoptosis in A549 cells by staining with propidium iodide and Annexin V-FITC. After treatment with its IC_50_ concentration, the acquired data showed a significant ability of **5a** to induce apoptosis and increase the early and late apoptotic cells by 3.77% and 19.89%, respectively (Figure 6). Interestingly, **5a** elevated the early and late apoptotic cells 77 and 9-folds compared to the negative control, respectively. These results implied that the hydantoin acetanilide derivatives arrested cells at the sub G1 and S phases and exhibited most of their antiproliferative activity by inducing cellular apoptosis and dissipating cellular integrity.

### 2.8. Wound Healing Assay Tau Aggregation Inhibition in a Cell Model of Tauopathy and Western Blot Analysis

Various articles have highlighted the role of EGFR in the migration of tumor cells [25]. Accordingly, the migratory effects of **5a** in A549 cells were assessed using a wound healing assay. Compound **5a** reduced the wound closure and the migration of A549 cells substantially in a dose-dependent manner (Figure 7).

### 2.9. Molecular Docking Study

The docking study on the EGFR (PDB: 5GTY) showed that all the target compounds **5a**–**5n** could interact with the active sites of the co-crystallized ligand binding sites with Met790 by hydrogen bonds [24]. Moreover, all compounds showed a hydrogen bonding interaction with Thr854 while the formed pi-H interaction with Leu777 in addition to hydrogen bonding with Lys745. The most potent derivative **5a** formed two hydrogen bonds with the gatekeeper mutant Met790, whereas the p-methoxy group was in a higher proximity to Cys797 of the kinase hinge (Figure 8). Conversely, erlotinib showed no binding interaction with Met790 which may explain the superiority of the target compound in the in vitro inhibition of the mutated EGFR growth (Appendix A). The target compounds showed docking scores (kcal/mole) ranging from −7.9294 to −9.2064 while gefitinib (Appendix A), erlotinib, and ATP showed −9.5254, −9.5245, and −9.54906, respectively. Interestingly, **5a** approximately occupied the EGFR pocket the same as the native ligand, and ATP (Figure 9).

## 3. Materials and Methods

### 3.1. Chemistry

Chemicals were purchased from Aldrich (Burlington, MA, USA), Merck (Darmstadt, Germany), and El-Nasr pharmaceutical chemicals companies (Qalyubia, Egypt), and used without further purification. Reactions were monitored by thin layer chromatography (TLC), using Merck9385 pre-coated aluminum plate silica gel (Kieselgel 60) 5 × 20 cm plates with a layer thickness of 0.2 mm. The spots were detected by exposure to a UV lamp at 254 nm. Melting points were determined on Stuart electro thermal melting point apparatus and were uncorrected. NMR spectra were carried out using a Bruker Avance 400 MHz ^1^HNMR spectrometer and 100 MHz ^13^CNMR spectrometer (Beni Sweif, Egypt), using TMS as internal reference. Chemical shifts (δ values are given in parts per million (ppm) relative to TMS DMSO-*d*_6_ (2.5 and 3.5 ppm for ^1^HNMR and 76.9 ppm for ^13^CNMR) and coupling constants (J) were measured in Hertz. Splitting patterns were designated as follows: s, singlet; d, doublet; t, triplet; q, quartet; m, multiplet. Elemental analyses were recorded on Shimadzu GC/MS-QP5050A, Regional center for Mycology and Biotechnology, Al-Azhar University, Cairo, Egypt. Mass spectra were recorded on an Advion compact mass spectrometer (CMS) and reported as mass/charge (*m*/*z*), Nawah scientific center for research, Almokattam, Cairo, Egypt. HPLC analysis of compound **5a** was performed on a Waters 2690 Alliance HPLC system equipped with a Waters 996 photodiode array detector, and UV detection at λ_max_ 254 nm, and HPLC run data were processed at Nawah Scientific, Egypt.

### 3.2. General Procedure for the Synthesis of Compounds ***2a–2n***


The appropriate aniline **1a**–**1n** (1 mmol) was dissolved in DCM, TEA (2 mmol) was added, and the reaction mixture was stirred in an ice bath. Chloroacetyl chloride (2 mmol) was added dropwise to the reaction mixture. The progress of the reaction was monitored by TLC. After the reaction was completed, the organic solvent evaporated, and the product was washed with distilled water.

### 3.3. General Procedure for the Synthesis of Compound ***4***

In a stoppered flask, acetophenone (1 mmol) was dissolved in a mixture of ethanol and distilled water (1:1). Potassium cyanide (3 mmol) and anhydrous ammonium carbonate (7 mmol) were carefully added. The mixture was heated in an oil bath at 58–61 °C for 48 h. After the reaction was completed, the mixture was concentrated under a vacuum to the third of its initial volume and diluted with water and acidified with diluted hydrochloric acid. The precipitate formed was filtered off, washed with water, and recrystallized from water ethanol mixture.

General procedure for the synthesis of compounds **5a**–**5****n** was as follows.

The hydantoin 4 (190, 1 mmol) was stirred with K_2_CO_3_ (10 mmol) in CH_3_CN for 30 min. The appropriate acylated aniline derivative **2a**–**2****n** (1 mmol) was then added portion wise then KI (10 mmol) was added. The mixture was heated at reflux for 3–6 h. The reaction progress was monitored by TLC, then concentrated under reduced pressure. The residue was triturated with ethyl acetate (3 × 30 mL) and the combined organic phase was washed with water (3 × 30 mL), dried over anhydrous sodium sulphate, and evaporated to dryness. The obtained product was recrystallized from a mixture of ethyl acetate and petroleum ether. 


**
*N*
**
**-(3,4-dimethoxyphenyl)-2-(4-methyl-2,5-dioxo-4-phenylimidazolidin-1-yl)acetamide (5a).**


Pale pink powder (165 mg, 43% yield); mp 145–148 °C; ^1^HNMR (400 MHz, DMSO-*d*_6_) δ (ppm): 10.11 (s, 1H, CO-NH-Ar), 8.96 (s, 1H, CO-NH), 7.5–7.49 (m, 2H, Ar-H), 7.40 (t, *J* = 7.5 Hz, 2H, Ar-H), 7.33 (t, *J* = 7.2 Hz, 1H, Ar-H), 7.21 (d, *J* = 2.3 Hz, 1H, Ar-H), 7.02 (dd, *J* = 8.7, 2.3 Hz, 1H, Ar-H), 6.86 (d, *J* = 8.7 Hz, 1H, Ar-H), 4.15 (s, 2H, CO-CH_2_-N), 3.69 (s, 3H, O-CH_3_), 3.69 (s, 3H, O-CH_3_), 1.71 (s, 3H, CH_3_); ^13^CNMR (100 MHz, DMSO-*d*_6_) δ (ppm): 175.36 (C=O), 164.17 (C=O), 155.36 (C=O), 148.60 (C quaternary), 144.99 (C quaternary), 139.51 (C quaternary), 132.18 (C quaternary), 128.52 (CH), 128.01 (CH), 125.68 (CH), 112.10 (CH), 110.98 (CH), 104.18 (CH), 63.12 (C quaternary), 55.71 (O-CH_3_), 55.39 (O-CH_3_), 40.67 (CH_2_), 24.96 (CH_3_); ESI/MS: *m*/*z* Calcd. for [M + Na]^+^: 406.1, Found: 405.8; Anal. Calcd for C_20_H_21_N_3_O_5_ (383.1): C, 62.65; H, 5.52; N, 10.96. Found: C, 62.93; H, 5.80; N, 11.23; HPLC-analysis: gradient elution H_2_O-MeOH (90:10) for 20 min, followed by with H_2_O-MeOH (20:80) for 5 min, and finally with a further isocratic elution with H_2_O-MeOH (90:10) for 5 min at a flow rate 1 ml/min, Rt = 19.947 min.


***N*-(4-aminosulphonylphenyl)-2-(4-methyl-2,5-dioxo-4-phenylimidazolidin-1-yl)acetamide (5b).**


Pale yellow powder (137 mg, 35% yield); mp 122–125 °C; ^1^HNMR (400 MHz, DMSO-*d*_6_) δ (ppm):10.60 (s, 1H, CO-NH-Ar), 9.01 (s, 1H, CO-NH), 7.57–7.47 (m, 4H, Ar-H), 7.45–7.31 (m, 5H, Ar-H), 4.24 (s, 2H, CO-CH_2_-N), 2.87 (s, 1H, SO_2_NH), 2.72 (s, 1H, SO_2_NH), 1.73 (s, 3H, CH_3_); ^13^CNMR (100 MHz, DMSO-*d*_6_) δ (ppm): 175.32 (C=O), 165.13 (C=O), 155.27 (C=O), 144.74 (C quaternary), 139.44 (C quaternary), 138.88 (C quaternary), 129.61 (CH), 128.53 (CH), 128.02 (CH), 125.66 (CH), 121.92 (CH), 120.61 (CH), 116.08 (CH), 63.16 (C quaternary), 40.79 (CH_2_), 24.87 (CH_3_); ESI/MS: *m*/*z* Calcd. for [M + Na]^+^: 425.1, Found: 424.8; Anal. Calcd for C_18_H_18_N_4_O_5_S (402.1): C, 53.72; H, 4.51; N, 13.92; S, 7.97. Found: C, 53.94; H, 4.59; N, 13.75; S, 8.04.


***N*-(3-chloro-4-methoxyphenyl)-2-(4-methyl-2,5-dioxo-4-phenylimidazolidin-1-yl)acetamide (5c).**


Pale black powder (192 mg, 50% yield); mp 163–166 °C; ^1^HNMR (400 MHz, DMSO-*d*_6_) δ (ppm):10.29 (s, 1H, CO-NH-Ar), 8.98 (s, 1H, CO-NH), 7.71 (d, *J* = 2.5 *Hz*, 1H, Ar-H), 7.55–7.49 (m, 2H, Ar-H), 7.44–7.30 (m, 4H, Ar-H), 7.09 (d, J = 9.0 *Hz*, 1H, Ar-H), 4.18 (s, 2H, CO-CH_2_-N), 3.80 (s, 3H, O-CH_3_), 1.72 (s, 3H, CH_3_); ^13^CNMR (100 MHz, DMSO-*d*_6_) δ (ppm): 175.33 (C=O), 164.54 (C=O), 155.31 (C=O), 150.64 (C quaternary), 139.46 (C quaternary), 132.23 (C quaternary), 128.52 (C quaternary), 128.00 (CH), 125.67 (CH), 125.32 (CH), 120.66 (CH), 118.91 (CH), 113.02 (CH), 63.13 (C quaternary), 56.16 (O-CH_3_), 40.66 (CH_2_), 24.84 (CH_3_); ESI/MS: *m*/*z* Calcd. for [M + Na]^+^: 410.1, Found: 410.0; Anal. Calcd for C_19_H_18_ClN_3_O_4_ (387.1): C, 58.84; H, 4.68; N, 10.84. Found: C, 58.95; H, 4.80; N, 11.02.


***N*-(2-methoxy-5-methylphenyl)-2-(4-methyl-2,5-dioxo-4-phenylimidazolidin-1-yl)acetamide (5d).**


Pale yellow powder (203 mg, 55% yield); mp 170–173 °C; ^1^HNMR (400 MHz, DMSO-*d*_6_) δ (ppm): 9.48 (s, 1H, CO-NH-Ar), 8.97 (s, 1H, CO-NH), 7.75 (s, 1H, Ar-H), 7.55–7.50 (m, 2H, Ar-H), 7.41 (t, *J* = 7.4 Hz, 2H, Ar-H), 7.36–7.31 (m, 1H, Ar-H), 6.93–6.85 (m, 2H, Ar-H), 4.27 (s, 2H, CO-CH_2_-N), 3.80 (s, 3H, O-CH_3_), 2.20 (s, 3H, Ar-CH_3_), 1.72 (s, 3H, CH_3_); ^13^CNMR (100 MHz, DMSO-*d*_6_) δ (ppm): 175.37 (C=O), 164.79 (C=O), 155.36 (C=O), 147.24 (C quaternary), 139.50 (C quaternary), 129.02 (C quaternary), 128.51 (C quaternary), 127.97 (CH), 126.50 (CH), 125.65 (CH), 124.65 (CH), 122.12 (CH), 111.06 (CH), 63.08 (C quaternary), 55.75 (O-CH_3_), 40.81 (CH_2_), 24.90 (CH_3_), 20.42 (CH_3_); ESI/MS: *m*/*z* Calcd. for [M + Na]^+^: 390.2, Found: 390.0; Anal. Calcd for C_20_H_21_N_3_O_4_ (367.2): C, 65.38; H, 5.76; N, 11.44. Found: C, 65.46; H, 5.94; N, 11.65.


***N*-(4-fluorophenyl)-2-(4-methyl-2,5-dioxo-4-phenylimidazolidin-1-yl)acetamide (5e).**


Pale gray powder (120 mg, 35% yield); mp 226–228 °C; ^1^HNMR (400 MHz, DMSO-*d*_6_) δ (ppm): 10.33 (s, 1H, CO-NH-Ar), 8.97 (s, 1H, CO-NH), 7.58–7.49 (m, 4H, Ar-H), 7.40 (t, *J* = 7.5 Hz, 2H, Ar-H), 7.35–7.30 (m, 1H, Ar-H), 7.13 (t, *J* = 8.9 Hz, 2H, Ar-H), 4.19 (s, 2H, CO-CH_2_-N), 1.71 (s, 3H, CH_3_); ^13^CNMR (100 MHz, DMSO-*d*_6_) δ (ppm): 175.37 (C=O), 164.62 (C=O), 159.30 (C=O), 155.33 (C quaternary), 139.47 (C quaternary), 134.95 (C quaternary), 128.51 (CH), 128.01 (CH), 125.69 (CH), 120.81 (CH), 115.32 (CH), 63.13 (C quaternary), 40.69 (CH_2_), 24.79 (CH_3_); ESI/MS: *m*/*z* Calcd. for [M + Na]^+^: 364.1, Found: 364.0; Anal. Calcd for C_18_H_16_FN_3_O_3_ (341.1): C, 63.34; H, 4.72; N, 12.31. Found: C, 63.57; H, 4.79; N, 12.39.


***N*-(2-fluorophenyl)-2-(4-methyl-2,5-dioxo-4-phenylimidazolidin-1-yl)acetamide (5f).**


Pale yellow powder (100 mg, 30% yield); mp 153–155 °C; ^1^HNMR (400 MHz, DMSO-*d*_6_) δ (ppm): 10.14 (s, 1H, CO-NH-Ar), 8.99 (s, 1H, CO-NH), 7.94–7.84 (m, 1H, Ar-H), 7.55–7.52 (m, 2H, Ar-H), 7.41 (t, *J* = 7.5 Hz, 2H, Ar-H), 7.38–7.31 (m, 1H, Ar-H), 7.30–7.23 (m, 1H, Ar-H), 7.21–7.11 (m, 2H, Ar-H), 4.30 (s, 2H, CO-CH_2_-N), 1.73 (s, 3H, CH_3_); ^13^CNMR (100 MHz, DMSO-*d*_6_) δ (ppm): 175.36 (C=O), 165.28 (C=O), 155.30 (C=O), 154.49 (C quaternary), 152.06 (C quaternary), 139.46 (C quaternary), 128.51 (CH), 128.00 (CH), 125.68 (CH), 124.47 (CH), 123.64 (CH), 115.63 (CH), 115.44 (CH), 63.12 (C quaternary), 40.62 (CH_2_), 24.76 (CH_3_); ESI/MS: *m*/*z* Calcd. for [M + Na]^+^: 364.1, Found: 363.9; Anal. Calcd for C_18_H_16_FN_3_O_3_ (341.1): C, 63.34; H, 4.72; N, 12.31. Found: C, 63.50; H, 4.88; N, 12.42.


***N*-(3-fluorophenyl)-2-(4-methyl-2,5-dioxo-4-phenylimidazolidin-1-yl)acetamide (5g).**


Pale yellow powder (240 mg, 70% yield); mp 182–185 °C; ^1^HNMR (400 MHz, DMSO-*d*_6_) δ (ppm): 10.52 (s, 1H, CO-NH-Ar), 8.99 (s, 1H, CO-NH), 7.55–7.47 (m, 3H, Ar-H), 7.40 (t, *J* = 7.5 Hz, 2H, Ar-H), 7.35–7.30 (m, 2H, Ar-H), 7.25 (d, *J* = 8.2 Hz, 1H, Ar-H), 6.91–6.84 (m, 1H, Ar-H), 4.21 (s, 2H, CO-CH_2_-N), 1.71 (s, 3H, CH_3_); ^13^CNMR (100 MHz, DMSO-*d*_6_) δ (ppm): 175.33 (C=O), 165.11 (C=O), 163.34 (C=O), 160.94 (C quaternary), 155.28 (C quaternary), 139.43 (C quaternary), 130.50 (CH), 128.52 (CH), 128.02 (CH), 125.67 (CH), 114.84 (CH), 110.12 (CH), 106.03 (CH), 63.15 (C quaternary), 40.81 (CH_2_), 24.83 (CH_3_); ESI/MS: *m*/*z* Calcd. for [M + Na]^+^: 364.1, Found: 363.9; Anal. Calcd for C_18_H_16_FN_3_O_3_ (341.1): C, 63.34; H, 4.72; N, 12.31. Found: C, 63.49; H, 4.86; N, 12.34.


***N*-(3,4-difluorophenyl)-2-(4-methyl-2,5-dioxo-4-phenylimidazolidin-1-yl)acetamide (5h).**


Pale yellow powder (110 mg, 31% yield); mp 195–198 °C; ^1^HNMR (400 MHz, DMSO-*d*_6_) δ (ppm): 10.71 (s, 1H, CO-NH-Ar), 9.03 (s, 1H, CO-NH), 7.57–7.51 (m, 2H, Ar-H), 7.42 (t, *J* = 7.5 Hz, 2H, Ar-H), 7.38–7.32 (m, 1H, Ar-H), 7.29–7.22 (m, 2H, Ar-H), 6.98–6.88 (m, 1H, Ar-H), 4.24 (s, 2H, CO-CH_2_-N), 1.74 (s, 3H, CH_3_); ^13^CNMR (100 MHz, DMSO-*d*_6_) δ (ppm): 175.27 (C=O), 165.49 (C=O), 163.61 (C=O), 161.35 (C quaternary), 155.20 (C quaternary), 140.95 (C quaternary), 139.40 (C quaternary), 128.53 (CH), 128.03 (CH), 125.65 (CH), 102.15 (CH), 101.86 (CH), 98.75 (CH), 63.17 (C quaternary), 40.86 (CH_2_), 24.85 (CH_3_); ESI/MS: *m*/*z* Calcd. for [M + Na]^+^: 382.1, Found: 381.8; Anal. Calcd for C_18_H_1__5_F_2_N_3_O_3_ (359.1): C, 60.17; H, 4.21; N, 11.69. Found: C, 60.41; H, 4.37; N, 11.88.


***N*-(2,3,4-trifluorophenyl)-2-(4-methyl-2,5-dioxo-4-phenylimidazolidin-1-yl)acetamide (5i).**


Yellow powder (110 mg, 30% yield); mp 205–210 °C; ^1^HNMR (400 MHz, DMSO-*d*_6_) δ (ppm): 10.36 (s, 1H, CO-NH-Ar), 9.00 (s, 1H, CO-NH), 7.69–7.59 (m, 1H, Ar-H), 7.56–7.49 (m, 2H, Ar-H), 7.41 (t, *J* = 7.4 Hz, 2H, Ar-H), 7.34 (t, *J* = 7.3 Hz, 1H, Ar-H), 7.31–7.24 (m, 1H, Ar-H), 4.29 (s, 2H, CO-CH_2_-N), 1.72 (s, 3H, CH_3_); ^13^CNMR (100 MHz, DMSO-*d*_6_) δ (ppm): 175.30 (C=O), 165.55 (C=O), 155.21 (C=O), 139.41 (C quaternary), 128.51 (C quaternary), 128.01 (C quaternary), 127.81 (C quaternary), 125.66 (C quaternary), 125.32 (CH), 123.55 (CH), 118.42 (CH), 111.96 (CH), 111.78 (CH), 63.13 (C quaternary), 40.52 (CH_2_), 24.73 (CH_3_); ESI/MS: *m*/*z* Calcd. for [M + Na]^+^: 400.1, Found: 399.9; Anal. Calcd for C_18_H_1__4_F_3_N_3_O_3_ (377.1): C, 57.30; H, 3.74; N, 11.14. Found: C, 57.46; H, 3.90; N, 11.30.


***N*-(4-methoxy-2-methylphenyl)-2-(4-methyl-2,5-dioxo-4-phenylimidazolidin-1-yl)acetamide (5j).**


Pale yellow powder (112 mg, 30% yield); mp 143-146 ℃; ^1^HNMR (400 MHz, DMSO-*d*_6_) δ (ppm): 9.48 (s, 1H, CO-NH-Ar), 8.92 (s, 1H, CO-NH), 7.50 (d, *J* = 7.4 Hz, 2H, Ar-H), 7.38 (t, *J* = 7.4 Hz, 2H, Ar-H), 7.31 (t, *J* = 7.2 *Hz*, 1H, Ar-H), 7.17 (d, *J* = 8.7 *Hz*, 1H, Ar-H), 6.77 (d, *J* = 2.7 Hz, 1H, Ar-H), 6.70 (dd, *J* = 8.7, 2.7 Hz, 1H, Ar-H), 4.19 (s, 2H, CO-CH_2_-N), 3.69 (s, 3H, O-CH_3_), 2.13 (s, 3H, Ar-CH_3_), 1.69 (s, 3H, CH_3_); ^13^CNMR (100 MHz, DMSO-*d*_6_) δ (ppm): 175.44 (C=O), 164.93 (C=O), 156.90 (C=O), 155.42 (C quaternary), 139.55 (C quaternary), 133.83 (C quaternary), 128.63 (C quaternary), 128.49 (CH), 127.99 (CH), 126.52 (CH), 125.73 (CH), 115.42 (CH), 111.24 (CH), 63.08 (C quaternary), 55.16 (O-CH_3_), 40.42 (CH_2_), 24.72 (CH_3_), 17.97 (CH_3_); ESI/MS: *m*/*z* Calcd. for [M + Na]^+^: 390.2, Found: 390.1; Anal. Calcd for C_20_H_21_N_3_O_4_ (367.2): C, 65.38; H, 5.76; N, 11.44. Found: C, 65.49; H, 5.84; N, 11.57.


***N*-(3-chloro-4-fluorophenyl)-2-(4-methyl-2,5-dioxo-4-phenylimidazolidin-1-yl)acetamide (5k).**


Pale yellow powder (195 mg, 55% yield); mp 170–173 °C; ^1^HNMR (400 MHz, DMSO-*d*_6_) δ (ppm): 10.53 (s, 1H, CO-NH-Ar), 9.01 (s, 1H, CO-NH), 7.86 (dd, *J* = 6.8, 2.3 Hz, 1H, Ar-H), 7.56–7.51 (m, 2H, Ar-H), 7.46–7.39 (m, 3H, Ar-H), 7.38–7.32 (m, 2H, Ar-H), 4.22 (s, 2H, CO-CH_2_-N), 1.73 (s, 3H, CH_3_); ^13^CNMR (100 MHz, DMSO-*d*_6_) δ (ppm): 175.30 (C=O), 165.05 (C=O), 155.24 (C=O), 154.43 (C quaternary), 152.01 (C quaternary), 139.42 (C quaternary), 135.74 (C quaternary), 128.52 (CH), 128.01 (CH), 125.65 (CH), 120.48 (CH), 119.36 (CH), 116.99 (CH), 63.15 (C quaternary), 40.72 (CH_2_), 24.83 (CH_3_); ESI/MS: *m*/*z* Calcd. For [M + Na]^+^: 398.1, Found: 397.9; Anal. Calcd for C_18_H_15_ClFN_3_O_3_ (375.1): C, 57.53; H, 4.02; N, 11.18. Found: C, 57.75; H, 4.21; N, 11.40.


***N*-(4-ethoxyphenyl)-2-(4-methyl-2,5-dioxo-4-phenylimidazolidin-1-yl)acetamide (5l).**


Pale brown powder (225 mg, 62% yield); mp 153–155 °C; ^1^HNMR (400 MHz, DMSO-*d*_6_) δ (ppm): 10.12 (s, 1H, CO-NH-Ar), 8.97 (s, 1H, CO-NH), 7.56–7.51 (m, 2H, Ar-H), 7.47–7.39 (m, 4H, Ar-H), 7.35 (t, *J* = 7.3 Hz, 1H, Ar-H), 6.87 (d, *J* = 9.0 Hz, 2H, Ar-H), 4.18 (s, 2H, CO-CH_2_-N), 3.97 (q, J = 7.0 Hz, 2H, O-CH_2_CH_3_), 1.73 (s, 3H, -CH_3_), 1.30 (t, *J* = 7.0 Hz, 3H, -CH_3_CH_2_); ^13^CNMR (100 MHz, DMSO-*d*_6_) δ (ppm): 175.39 (C=O), 164.10 (C=O), 155.38 (C=O), 154.61 (C quaternary), 139.50 (C quaternary), 131.63 (C quaternary), 128.50 (CH), 127.98 (CH), 125.70 (CH), 120.56 (CH), 114.47 (CH), 63.09 (C quaternary), 63.09 (O-CH_2_), 40.63 (CH_2_), 24.80 (CH_3_), 14.68 (CH_3_); ESI/MS: *m*/*z* Calcd. for [M + Na]^+^: 390.2, Found: 389.9; Anal. Calcd for C_20_H_21_N_3_O_4_ (367.2): C, 65.38; H, 5.76; N, 11.44. Found: C, 65.46; H, 5.89; N, 11.56.


***N*-(2-methoxyphenyl)-2-(4-methyl-2,5-dioxo-4-phenylimidazolidin-1-yl)acetamide (5m).**


Yellow powder (160 mg, 45% yield); mp 85–90 °C; ^1^HNMR (400 MHz, DMSO-*d*_6_) δ (ppm): 9.56 (s, 1H, CO-NH-Ar), 8.98 (s, 1H, CO-NH), 7.93 (d, *J* = 7.8 Hz, 1H, Ar-H), 7.57–7.50 (m, 2H, Ar-H), 7.42 (t, *J* = 7.4 Hz, 2H, Ar-H), 7.34 (t, *J* = 7.3 Hz, 1H, Ar-H), 7.11–7.02 (m, 2H, Ar-H), 6.92–6.87 (m, 1H, Ar-H), 4.30 (s, 2H, CO-CH_2_-N), 3.85 (s, 3H, O-CH_3_), 1.73 (s, 3H, -CH_3_); ^13^CNMR (100 MHz, DMSO-*d*_6_) δ (ppm): 175.42 (C=O), 164.89 (C=O), 155.38 (C=O), 149.31 (C quaternary), 139.49 (C quaternary), 128.50 (C quaternary), 127.99 (CH), 126.79 (CH), 125.68 (CH), 124.53 (CH), 121.53 (CH), 120.27 (CH), 111.21 (CH), 63.10 (C quaternary), 55.69 (O-CH_3_), 40.85 (CH_2_), 24.79 (CH_3_); ESI/MS: *m*/*z* Calcd. for [M + Na]^+^: 376.1, Found: 375.9; Anal. Calcd for C_19_H_19_N_3_O_4_ (353.1): C, 64.58; H, 5.42; N, 11.89. Found: C, 64.31; H, 5.31; N, 11.70.


***N*-(3-methoxyphenyl)-2-(4-methyl-2,5-dioxo-4-phenylimidazolidin-1-yl)acetamide (5n).**


Pale yellow powder (230 mg, 65% yield); mp 148–151 °C; ^1^HNMR (400 MHz, DMSO-*d*_6_) δ (ppm): 10.25 (s, 1H, CO-NH-Ar), 8.98 (s, 1H, CO-NH), 7.55–7.49 (m, 2H, Ar-H), 7.40 (t, *J* =7.5 Hz, 2H, Ar-H), 7.33 (t, *J* = 7.2 Hz, 1H, Ar-H), 7.22–7.20 (m, 1H, Ar-H), 7.20–7.17 (m, 1H, Ar-H), 7.06 (d, *J* = 8.0 Hz, 1H, Ar-H), 6.63 (dd, *J* = 8.1, 2.2 Hz, 1H, Ar-H), 4.18 (s, 2H, CO-CH_2_-N), 3.70 (s, 3H, O-CH_3_), 1.71 (s, 3H, -CH_3_); ^13^CNMR (100 MHz, DMSO-*d*_6_) δ (ppm): 175.36 (C=O), 164.71 (C=O), 159.58 (C=O), 155.34 (C quaternary), 139.73 (C quaternary), 139.48 (C quaternary), 129.69 (CH), 128.52 (CH), 128.01 (CH), 125.68 (CH), 111.33 (CH), 109.01 (CH), 104.82 (CH), 63.14 (C quaternary), 55.01 (O-CH_3_), 40.80 (CH_2_), 24.88 (CH_3_); ESI/MS: *m*/*z* Calcd. for [M + Na]^+^: 376.1, Found: 375.9; Anal. Calcd for C_19_H_19_N_3_O_4_ (353.1): C, 64.58; H, 5.42; N, 11.89. Found: C, 64.79; H, 5.56; N, 11.96.

### 3.4. Anti-Proliferative Activities against A549, H1975, PC9, WI-38 Cell Lines

The new prepared candidates **5a**–**5n** were evaluated for their antitumor activities against four selected cell lines, the adenocarcinomic human alveolar basal epithelial cells NCI-A549, NCI-H1975, human pulmonary adenocarcinoma PC9, and the caucasian fibroblast-like fetal lung cells WI-38 (purchased from American Type Culture Collection (ATCC), United States of America), using an MTT assay [26], as described earlier [27].

### 3.5. EGFR Inhibition Assay 

The EGFR inhibitory assay was performed using EGFR (wild, L858R, or T790M) Kinase Assay Kits (Cat. # 40321, 40324, or 40323, respectively, BPS Bioscience, San Diego, CA, USA) as described earlier [28,29].

### 3.6. In Vitro Inhibition of EGFR Downstream Pathway 

#### 3.6.1. Akt Enzyme Assay

Akt expression level was assayed using the Phospho Sandwich Akt (Phospho-Ser473) ELISA Kit (Cat. # MBS9511022, MyBioSource, San Diego, CA, USA) containing the components necessary for the semi-quantitative determination of Akt concentrations within experimental cell lysate samples. The assay was performed according to company protocols: the lysates were diluted with assay diluent and 100 μL/well was dispensed, wrapped with parafilm, and incubated for 2 h. They were shaken with RT or overnight at 4 °C. The plate was washed 3 times with 1× Wash Buffer 300 μL/well with a vacuum-based plate washer. Then, the plate was stricken on absorbent paper to remove as much residual liquid as possible. The Detection Antibody was diluted to 1× and 100 μL/well was dispensed, wrapped with parafilm, and incubated on a shaker for 2 hrs at RT, and the washing step was repeated. Streptavidin-HRP was diluted to 1× and 100 μL/well was dispensed, wrapped with parafilm, and incubated on a shaker for 30 min at RT. Ready-to-use substrate was brought to RT and 100 μL/well was dispensed and the reaction was allowed to incubate on an orbital shaker for 20–30 min. The reaction was stopped with stop solution 100 μL/well. The wells turned from blue to yellow and were read at 450 nm.

#### 3.6.2. ERK Enzyme Assay

ERK ELISA Kit (Cat. # E4317-100, BioVision, Inc., Waltham, MA, USA) was used for the in vitro quantitative determination of Human ERK. Firstly, the plate was washed 2 times with Wash Solution before the standard, sample, and control wells were added. A total of 100 μL of each standard or sample was added into appropriate wells. It was covered well and incubated for 1.5 h at 37 °C. The cover was removed and the plate content was discarded without washing or letting the wells completely dry. A total of 0.1 mL of Biotin-detection antibody work solution was added into the above wells. The plate was sealed and incubated at 37 °C for 60 min. The solution was discarded and washed 3 times with Wash Solution. It was washed by filling each well with Wash Buffer (350 μL) using a multi-channel pipette or auto-washer. It was soaked for 1–2 min, and then all residual wash-liquid was removed from the wells by aspiration. After the last wash, any remaining wash buffer was removed by aspirating or decanting. The plate was clapped on absorbent filter papers or other absorbent materials. A total of 0.1 mL of SABC working solution was added into each well, and the plate was covered and incubated at 37 °C for 30 min. The solution was discarded and washed 5 times with wash solution as mentioned before. A total of 90 μL of TMB substrate was added into each well, the plate was covered and incubated at 37 °C in the dark within 15–30 min. The shades of blue were seen in the first 3–4 wells by the end of the incubation. A total of 50 μL of Stop Solution was added to each well. The result was read at 450 nm within 20 min. 

### 3.7. Inhibition of EGFR Phosphorylation Expression Using Western Blot Analysis

The western blot experiment was performed according to the same procedure described earlier [30].

### 3.8. Cell Cycle Analysis and Detection of Apoptosis

Cell cycle analysis and Annexin V-FITC apoptosis assay were performed as described earlier [26].

### 3.9. Wound Healing Assay

Wound healing assay of **5a** was performed using CytoSelect™ 24-well Wound Healing Assay kit (Cat. # CBA-120, Cell Biolabs Inc., Cambridge, UK) per manufacturer’s instructions. In brief, the 24-well plate wound healing inserts were allowed to warm up at room temperature for 10 min., then to each well, 500 µL of A549 cells suspension (1.0 × 106) in media containing 10% fetal bovine serum (FBS) was added. Cells were kept in a cell culture incubator until a monolayer formed. Carefully, the insert was removed from the well. The media were slowly aspirated and discarded from the wells. The wells were washed with media to remove dead cells and debris. Finally, the media were added to wells to keep cells hydrated, then wells were visualized under a light microscope. Media containing the indicated concentrations of compound **5a** (0.5, 1, or 2 µM) were then added into the wells for 72 h. the wound closure was monitored with a light microscope. The percent closure of the cells into the wound field was measured.

### 3.10. Molecular Docking to EGFR Active Site 

All processes of docking algorithms and visualization were performed within the EGFR active site, using Molecular Operating Environment (MOE) 2019.01 (Chemical Computing Group, Montreal, QC, Canada) software (PDB: 5GTY) [31], downloaded from the Protein Data Bank [32]. The protein structure (PDB: 5GTY) was prepared through the QuickPrep suite executed in MOE. Then, all the structures of the docked hydantoin acetanilides were built, and their energies were minimized using the default force field, AMBER10. The native ligand was re-docked into EGFR active site to validate the docking study at the binding site using the default parameters. The top ranked re-docked pose had energy score of −10.82 kcal/mol and an RMSD value of 1.3070 Å. The acetanilides were docked in the EGFR active site using rigid receptor docking protocol, where ligand placement was performed with Triangle Matcher method using AMBER10 as a default force field and eventually ranking the docking poses by the London dG scoring function. The generated poses of the best binding affinities were considered. 

## 4. Conclusions 

A series of hydantoin acetanilide derivatives **5a**–**5n** were designed and synthesized as potent inhibitors of EGFR^L858R/T790M^. The synthesized derivatives exhibited potent activity on NSCLC tumor subpanels A549, H1975, and PC9 cells. Interestingly, compounds **5a** and **5f** showed a strong ability to EGFR with better affinity to the mutations L858R/T790M than the wild type and were safer than the reference erlotinib on the normal cells WI-38. Western blotting revealed a significant decrease in the expression of EGFR in A549 cells upon the treatment of **5a** at its IC_50_ value. Besides, **5a** inhibited EGFR related downstream proteins (ERK and Akt), induced cellular apoptosis, and arrested sub G1 and S phases of the cell cycle. The wound healing assay revealed the substantial ability of **5a** to decrease A549 cells’ migration in a dose-dependent manner. Eventually, it is conceivable that further in vivo investigation will be of significant interest to afford promising recipes to overcome the obstacle of EGFR resistance.

## Data Availability

The data presented in this study are available in article and Appendix A.

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
