# Peer review of "Design and Synthesis of New Hydantoin Acetanilide Derivatives as Anti-NSCLC Targeting EGFRL858R/T790M Mutations"

_pharmaceuticals, 2022, doi:10.3390/ph15070857_

Round 1

Reviewer 1 Report

The review titled  Design and synthesis of new hydantoin acetanilide derivatives 2

as anti-NSCLC targeting EGFRL858R/T790M mutations. after consideration of major comments.

.

1)      Abstract, a) the authors should add standard result and short conclusion  

2)      Introduction, the authors should add rational for his work as authors mentioned some drugs which are completely different in structure for example Erlotinib even that new compounds are somewhat isostere with Osimertinib

3)      Authors synthesized substituted aniline but cancel substitution at another phenyl ring, please explain.

4)      Results and discussion, authors did not discussion NMR fundings why?

5)      Authors test their compounds against A549, H1975, PC9, WI-38 but in the title mentioned NSCLC. why?

6)      Also, authors did not write full name for used cell lines

7)      The differences in activity against EGVRs should be rational.

8)      NMR peaks should assign to corresponding Carbon or proton.

Author Response

Dear/ Editor-In-Chef

Pharmaceuticals Journal

We are extremely grateful to the editor and the reviewers for the positive, insightful and constructive comments, which certainly helps to improve the quality of the manuscript. Herewith, we are submitting point to point response to comments posed by the reviewers. We have incorporated the suggestions/corrections made by the reviewers in the revised manuscript. The suggestions were very much helpful in shaping the manuscript. We are thankful to the reviewer for their valuable inputs. The point-to-point corrections and responses to comments are given as follows.

Reviewer #1: The review titled Design and synthesis of new hydantoin acetanilide derivatives 2

as anti-NSCLC targeting EGFRL858R/T790M mutations. after consideration of major comments.

1)      Abstract, a) the authors should add standard result and short conclusion  

Modification was done.

2)      Introduction, the authors should add rational for his work as authors mentioned some drugs which are completely different in structure for example Erlotinib even that new compounds are somewhat isostere with Osimertinib

A figure (2) was added to the manuscript containing hydantoin bearing compounds (The same skeleton in our study) with EGFR inhibitory activity.

3)      Authors synthesized substituted aniline but cancel substitution at another phenyl ring, please explain.

The authors would like to thank the reviewer for this important note. Actually, this work is the first exploration of the hydantoin acetanilide in our group. We will cover this point in our next proposed research.

4)      Results and discussion, authors did not discussion NMR fundings why?

The NMR findings were added. “1H NMR spectra of compounds 5a-5n showed the hydantoin methyl protons as a singlet integrated for 3 protons at approximately δ 1.69-1.74 ppm. The hydantoin amidic NH proton appeared as a singlet at δ 9.03-8.92 ppm, whereas the other amide proton appeared more downfield shifted as a singlet at 10.71-9.48 ppm due to the anisotropic effect of the benzene ring.”

5)      Authors test their compounds against A549, H1975, PC9, WI-38 but in the title mentioned NSCLC. why?

The anticancer activity was evaluated on A549, H1975, and PC9. These cell lines are non small cell lung cancer (NSCLC) cells, whereas the active hydantoin derivatives 5a and 5f were tested on the normal fetal lung WI-38 cell line to evaluate the safety of the compounds on normal cells.

6)      Also, authors did not write full name for used cell lines.

The full names were added to the experimental part (4.3.).

7)      The differences in activity against EGVRs should be rational.

The aim of this work is to prepare a potent mutated EGFR inhibitors which raises the affinity towards Met790. The mutation significantly boosts the affinity of EGFR to ATP, leading to lessening the ability of EGFR inhibitors to compete with ATP effectively. In this respect, the designed compounds contain a hydantoin group which interacts with Met790 in a manner like ribose of the agonist ATP, whereas the phenyl ring is to interact with Leu777 and Asp855 like the adenine group of the ATP. The carbonyl group of the hydantoin group interacts with Thr854 like that of gefitinib. The carbonyl of the acetamide group binds with Lys745 by hydrogen bonding like ATP but ATP is interacting with Lys745 by an ionic bond.

8)      NMR peaks should assign to corresponding Carbon or proton.

       The modifications were done as requested.

Reviewer 2 Report

This manuscript is very well written. It is clear and easy to understand. The abstract, introduction, results & discussion and references sections are well drafted with no changes required. In Addition, no errors have been detected in supplementary materials and data. However, subsection 3.2 and 3.3. requires minor revision. 

Recommended minor corrections:

Line 48: cell growth,  downstream - space should be removed after comma. 

Line 114: (average IC50= 2.57-9.29 µM) - space should be added after IC50

Line118: (average IC50= 7.19-14.54 µM) - space should be added after IC50.

Line 119: (average IC50= 35.82-91.39 µM) - space should be added after IC50.

Line 163: 2.3-folds better superior than the reference 

Line 237 to 242:

3.2. General procedure for the synthesis of compounds 2a-n 

The appropriate aniline 1a-n (1mmol) was dissolved in DCM, TEA (2mmol) was added and the reaction mixture was stirred in an ice bath. Chloroacetyl chloride (2mmol) was drop-wisely added to the reaction mixture. The progress of the reaction was monitored by TLC. After the reaction was completed, the organic solvent was evaporated, and the product was washed with distilled water.

Line 238: (1mmol) - space should be added after 1.

Line 238: (2mmol) - space should be added after 2.

Line 239: (2mmol) - space should be added after 2.

Line 234 to 250:

3.3. General procedure for the synthesis of compound 4 

In a stoppered flask, acetophenone (1 mmol) was dissolved in a mixture of ethanol and distilled water (1:1). Potassium cyanide (3 mmol) and anhydrous ammonium carbonate (7 mmol) were carefully added. The mixture was heated in an oil bath at 58-61°C for 48 h. After the reaction was completed, the mixture was concentrated under vacuum to the third of its initial volume and diluted with water and acidified with (dilute or concentrated ?) hydrochloric acid. Then hydrochloric acid was added till the solution is acidic. The precipitate formed is was filtered off, washed with water, and recrystallized from water ethanol mixture.

Line 412: 4oC space should be added after 4.

Line 425, 427 and 433: 37ospace should be added after 37.

Recommendation for minor revision:

Subsection 3.3. General procedure for the synthesis of compounds 5a-n (line 251 to 259) should be revised:

The hydantoin 4 (190, 1 mmol) was stirred with K2CO3 (10 mmol) in CH3CN for 30 min. The appropriate acylated aniline derivative 2a-n (1 mmol) was then added portion wise then KI (10 mmol) was added. The mixture was heated at reflux for 3-6 h. The reaction progress was monitored via by TLC, then the reaction mixture was concentrated under reduced pressure and the remaining mixture was extracted using ethyl acetate (3 X 30 mL) and washed with water (3 X 30 mL). The organic layer was collected and dried over anhydrous sodium sulphate, solvent was evaporated, and the obtained product was recrystallized from a mixture of ethyl acetate and petroleum ether. 

When acetonitrile was removed under reduced pressure a residue containing product and excess potassium carbonate was remained in the flask. Then the procedure described that the remaining mixture, (which it is a solid at this point) was extracted ? using ethyl acetate (3 X 30 mL) and washed with water (3 X 30 mL)???  

extracted or triturated?

The highlighted procedure may revised as below:

The reaction progress was monitored by TLC, then concentrated  under reduced pressure. The residue was triturated with ethyl acetate (3 X 30 mL) and the combined organic phase washed with water (3 X 30 mL), dried over anhydrous sodium sulphate and evaporated to dryness. The obtained product was recrystallized from a mixture of ethyl acetate and petroleum ether. 

Line 273 and 274: N-(4-sulphonylaminophenyl)-2-(4-methyl-2,5-dioxo-4-phenylimidazolidin-1-yl)acetamide (5b).  

The highlighted IUPAC name should be changed to 4-aminosulphonylphenyl

Author Response

Reviewer #2: This manuscript is very well written. It is clear and easy to understand. The abstract, introduction, results & discussion and references sections are well drafted with no changes required. In Addition, no errors have been detected in supplementary materials and data. However, subsection 3.2 and 3.3. requires minor revision. 

The authors would like to thank the reviewer for his nice words.

Recommended minor corrections:

Line 48: cell growth,  downstream - space should be removed after comma. 

The modification was done as recommended.

Line 114: (average IC50= 2.57-9.29 µM) - space should be added after IC50. 

The modification was done as recommended.

Line118: (average IC50= 7.19-14.54 µM) - space should be added after IC50.

The modification was done as recommended.

Line 119: (average IC50= 35.82-91.39 µM) - space should be added after IC50.

The modification was done as recommended.

Line 163: 2.3-folds better superior than the reference.

The modification was done as recommended.

Line 237 to 242:

3.2. General procedure for the synthesis of compounds 2a-n 

The appropriate aniline 1a-n (1mmol) was dissolved in DCM, TEA (2mmol) was added and the reaction mixture was stirred in an ice bath. Chloroacetyl chloride (2mmol) was drop-wisely added to the reaction mixture. The progress of the reaction was monitored by TLC. After the reaction was completed, the organic solvent was evaporated, and the product was washed with distilled water.

Line 238: (1mmol) - space should be added after 1.

The modification was done as recommended.

Line 238: (2mmol) - space should be added after 2.

The modification was done as recommended.

Line 239: (2mmol) - space should be added after 2.

The modification was done as recommended.

3.3. General procedure for the synthesis of compound 4 

In a stoppered flask, acetophenone (1 mmol) was dissolved in a mixture of ethanol and distilled water (1:1). Potassium cyanide (3 mmol) and anhydrous ammonium carbonate (7 mmol) were carefully added. The mixture was heated in an oil bath at 58-61°C for 48 h. After the reaction was completed, the mixture was concentrated under vacuum to the third of its initial volume and diluted with water and acidified with (dilute or concentrated ?) hydrochloric acid. Then hydrochloric acid was added till the solution is acidic. The precipitate formed is was filtered off, washed with water, and recrystallized from water ethanol mixture.

The modification was done

Line 412: 4oC space should be added after 4.

The modification was done.

Line 425, 427 and 433: 37oC space should be added after 37.

The modification was done.

Recommendation for minor revision:

Subsection 3.3. General procedure for the synthesis of compounds 5a-n (line 251 to 259) should be revised:

The hydantoin 4 (190, 1 mmol) was stirred with K2CO3 (10 mmol) in CH3CN for 30 min. The appropriate acylated aniline derivative 2a-n (1 mmol) was then added portion wise then KI (10 mmol) was added. The mixture was heated at reflux for 3-6 h. The reaction progress was monitored via by TLC, then the reaction mixture was concentrated under reduced pressure and the remaining mixture was extracted using ethyl acetate (3 X 30 mL) and washed with water (3 X 30 mL). The organic layer was collected and dried over anhydrous sodium sulphate, solvent was evaporated, and the obtained product was recrystallized from a mixture of ethyl acetate and petroleum ether.

The modification was done as recommended.

When acetonitrile was removed under reduced pressure a residue containing product and excess potassium carbonate was remained in the flask. Then the procedure described that the remaining mixture, (which it is a solid at this point) was extracted ? using ethyl acetate (3 X 30 mL) and washed with water (3 X 30 mL)???  

extracted or triturated?

The highlighted procedure may revised as below:

The reaction progress was monitored by TLC, then concentrated under reduced pressure. The residue was triturated with ethyl acetate (3 X 30 mL) and the combined organic phase washed with water (3 X 30 mL), dried over anhydrous sodium sulphate and evaporated to dryness. The obtained product was recrystallized from a mixture of ethyl acetate and petroleum ether.

The modification was done as recommended.

Line 273 and 274: N-(4-sulphonylaminophenyl)-2-(4-methyl-2,5-dioxo-4-phenylimidazolidin-1-yl)acetamide (5b).  

The highlighted IUPAC name should be changed to 4-aminosulphonylphenyl

The Modification was done.

Round 2

Reviewer 1 Report

the article is accepted in the present form as authors covered all comments